# An Overview of Social Media Apps and their Potential Role in Geospatial Research

**Innocensia Owuor \* and Hartwig H. Hochmair** 

Geomatics Program, Fort Lauderdale Research and Education Center, University of Florida, 3205 College Ave, Davie, FL 33314, USA; hhhochmair@ufl.edu
\*  Correspondence: innocensia.owuor@ufl.edu

**Abstract:** Social media apps provide analysts with a wide range of data to study behavioral aspects of our everyday lives and to answer societal questions. Although social media data analysis is booming, only a handful of prominent social media apps, such as Twitter, Foursquare/Swarm, Facebook, or LinkedIn are typically used for this purpose. However, there is a large selection of less known social media apps that go unnoticed in the scientific community. This paper reviews 110 social media apps and assesses their potential usability in geospatial research through providing metrics on selected characteristics. About half of the apps (57 out of 110) offer an Application Programming Interface (API) for data access, where rate limits, fee models, and type of spatial data available for download vary strongly between the different apps. To determine the current role and relevance of social media platforms that offer an API in academic research, a search for scientific papers on Google Scholar, the Association for Computing Machinery (ACM) Digital Library, and the Science Core Collection of the Web of Science (WoS) is conducted. This search revealed that Google Scholar returns the highest number of documents (Mean = 183,512) compared to ACM (Mean = 1895) and WoS (Mean = 1495), and that data and usage patterns from prominent social media apps are more frequently analyzed in research studies than those of less known apps. The WoS citation database was also used to generate lists of themes covered in academic publications that analyze the 57 social media platforms that offer an API. Results show that among these 57 platforms, for 26 apps at least some papers evolve around a geospatial discipline, such as Geography, Remote Sensing, Transportation, or Urban Planning. This analysis, therefore, connects apps with commonly used research themes, and together with tabulated API characteristics can help researchers to identify potentially suitable social media apps for their research. Word clouds generated from titles and abstracts of papers associated with the 57 platforms, grouped into seven thematic categories, show further refinement of topics addressed in the analysis of social media platforms. Considering various evaluation criteria, such as provision of geospatial data or the number (i.e., absence) of currently published research papers in connection with a social media platform, the study concludes that among the numerous social media apps available today, 17 less known apps deserve closer examination since they might be used to investigate previously underexplored research topics. It is hoped that this study can serve as a reference for the analysis of the social media landscape in the future.

**Keywords:** API; data access; fee models; Google Scholar; Web of Science; ACM Digital Library

---

## 1. Introduction

Numerous social media apps have evolved in the recent years and significantly enhanced information sharing and networking capabilities among their users. Undeniably, these apps have permeated many aspects of modern life in our society.

Largely aided by the proliferation of smartphones and their use today, social media apps are increasingly location-based, providing analysts with access to a wide range of shared spatial data, such as check-ins, geo-tagged images, video clips or text messages, or reviews of businesses and other localities. Based on these data, research studies provide valuable insights into spatio-temporal aspects of marketing, event detection, political campaigning, disaster management, migration, transport, natural resource management, human mobility, urban planning, tourism, epidemics, and communication [1–6].

Although hundreds of social media apps have been developed, only a handful of them are commonly used in research, including Twitter, Facebook, Flickr, Foursquare, YouTube, LinkedIn, or Yelp. The presence of user selection bias, based on age, gender, socio-economic status, etc., is a well-known phenomenon in social media use. For example, Twitter shows a bias towards male users and an under-sampling of Hispanic and African American users in certain regions of the United States [7]. User characteristics also vary between the different social media apps. For example, a survey conducted in the United States found that Snapchat and Instagram are used primarily by young people, while YouTube and Facebook are more widely used by older generations [8]. The same study showed that LinkedIn is more biased towards higher-income users (>$75,000 per year) than other apps such as WhatsApp or Snapchat. Popularity also varies per region. WeChat, for example, is popular in China [9] whereas Vkontakte is widely used in Russia [10]. In general, the country where an app has been developed draws also a strong user base for that app. Examples include Dronestagram (France) [11] or Mapillary (Sweden) [12]. Different apps vary in the topical focus and therefore attract different users. For instance, Doximity is popular with medical professionals whereas Dronestagram attracts drone pilots who like to share airborne imagery. Due to these many facets of user selection bias, analysis results drawn from a single app may be biased as well. In consequence, having access to data from a wider range of social media platforms that reach different user groups in different geographic regions might, therefore, help to reduce user selection bias and its effect on analysis results, if combined in joint data analysis.

Analyzing the prominence of different social media apps and existing access options to their data through application programming interfaces (APIs) provides an important knowledge base for researchers to potentially expand the range of social media apps used for spatio-temporal research endeavors. Using this and the need for a better understanding of the social media landscape as a motivation, this research reviews 110 social media apps and focuses on three objectives:

- Identification of social media apps with APIs and the provision of location data through the APIs;
- Determination of the number of research papers published between 1997 and April 2020 on Google Scholar, the Association for Computing Machinery (ACM) Digital Library, and the Web of Science (WoS) that are related to the different social media apps with APIs;
- Extraction of research themes in academic literature covered in WoS for social media apps with APIs.

This study's predominant aim is to bring to light less known apps that provide spatial data, which is central to geospatial research. Data extracted from these apps may offer researchers information that is unavailable from other platforms and can be potentially used to tackle new research questions and fill current research gaps. The characteristics provided about these apps in this study, e.g., API rate limits or general theme, can guide researchers towards a closer examination of specific apps that align with their research interests and that may have been previously gone unnoticed.

Whilst access to location data from social media apps has been beneficial in various ways, it has broached concerns among social media users about data protection and privacy especially in the aftermath of the Cambridge Analytica scandal [13]. Many policies that provide guidelines for safeguarding data privacy of social media users have been instituted by governments and social media companies alike in the recent past. Examples are the General Data Protection Regulation (GDPR) [14] and, more recently, the California Consumer Privacy Act (CCPA) [15]. These policies tend

to raise challenges for harvesting data from social media apps, as will be discussed for selected apps in more detail.

## 2. Literature Review

Social media is a group of highly interactive platforms that use mobile and Web 2.0-based technologies to facilitate the creation and exchange of user-generated content [16,17]. Social media apps can be classified by purpose and function, which include social networking, microblogging, blogging, photo sharing, or crowdsourcing [18]. One of the first social media apps whose features are mirrored by today's popular social networks, such as Facebook, was SixDegrees, which was launched in 1997 and shut down in 2001 [19]. The wide reach of social media benefits businesses [20], emergency and disaster management [21], and the tracking of epidemics and diseases [22,23]. Social scientists pioneered the use of social media data for ethnographic studies that give better insights about people's behaviors and opinions [24–26]. Currently, over 9000 tweets are sent out per second and over 1000 photos are uploaded on Instagram per second [27]. This abundance of generated data is critical for real-time monitoring especially in emergency management and crisis mapping [22]. Research pertaining to social media has been largely facilitated by data access through APIs [28]. The availability of APIs in platforms such as Wikipedia, Twitter, Facebook, or Foursquare led to an increase in data services, software tools for analysis as well as social media analytics platforms [29]. API documentation provide detailed information about the API functionality and a description of data retrieval [30]. APIs allow third parties, such as games or productivity apps, to leverage friendship connections through regulated, programmatic access to recorded connections between registered users [31]. Access to public data that is published on the web may require login credentials [32] in order to be able to enforce download limits and to oversee access to data. For this purpose, social media sites have instituted authentications using keys and tokens. For example, Flickr allows up to 3600 queries per hour [33] and Twitter has varying rules for rate limiting on its APIs, such as 15 calls every 15 min [34]. Rate limits prevent abuse of data access and enhance security although this stifles generativity (i.e., third party use of the social graph to create innovative products or new insights into data [31]). API rate limiting can be implemented through request queues, throttling API calls (i.e., disconnect client or reduce bandwidth), and the implementation of rate limit algorithms [35].

Because of the Facebook Cambridge Analytica data breach, some social media apps have deprecated certain API methods or shut them down completely to protect user data and control its use. For example, Facebook recently further restricted data accessibility [36] and the 500px API is now available through paid subscription only [37].

The important role of social media for scientific research becomes evident by the sheer number of papers analyzing data from social media platforms. Various review studies and meta-analyses provide an overview of how data extracted from different social media platforms are analyzed, and how social media apps are used in different contexts and environments. For example, one paper reviews literature on the use of social media in academia [18]. It distinguishes between several categories of social media use, including social networking, social data sharing, video, blogging, microblogging, wikis, rating, and reviewing. It reports that the percentage of scholars who use social networking apps (e.g., Facebook, LinkedIn) for professional purposes is much lower than the percentage of scholars who use it for personal reasons. It also points towards a large variability in usage of different platforms among scholars, with numbers ranging between 10% for Twitter, 46% for ResearchGate, and 55% for YouTube. Another systematic review analyzes social media use for public health communication among the general public, patients, and health professionals based on 98 original research studies [38]. These studies included a range of social media tools and apps, with Facebook, blogs, Twitter, and YouTube being the most often reported tools. A review of 279 research papers describes the emerging role of social media in tourism and hospitality, which has only a short history [39]. Fewer than 10 papers were published in 2007, followed by a strong increase in the annual number of papers published. Popular keywords identified in analyzed publications include

marketing, consumer behavior and user-generated content. A review of 48 publications that focused on the use of social media for the assessment of nature-based tourism showed that images from Flickr were most often used (36 papers), followed by Panoramio images (10 papers), and Instagram images (6 papers). Point location data collected from social media platforms were mentioned in 40 papers and temporal data in 12 papers [40].

Another study examines the role of social media, in particular, Facebook, as an educational tool in higher education [41], concluding that Facebook usage comes with benefits such as increased teacher-student and student-student interaction, improved performance, and the convenience of learning and higher engagement. A review paper [42] analyzed 412 articles on Facebook in the context of Social Science to answer various overarching questions, including: (a) Who is using Facebook? (analysis of users); (b) Why do people use Facebook? (motivation); (c) Why are people disclosing personal information on Facebook (privacy and information disclosure). An overview of social media use in college classrooms [43] concludes that LinkedIn is the most used social network by university faculty. Based on the review of 92 papers, another study analyzed which academic disciplines publish results from Twitter analysis, showing that 46% of papers come from Computer Science, followed by 30% from Information Science, and 7% from Earth- and Geoscience [44]. Furthermore, it showed that 46% of analyses fall into the category of event detection (disaster/health/disease/traffic management), 14% into social network analysis, and 13% focus on retrieving direct or indirect geolocation information from Twitter. A detailed list of publications in six leading tourism and hospitality journals that collected and analyzed online reviews from online platforms, such as Booking.com, Ctrip, Expedia, TripAdvisor, or Yelp, to address a variety of research questions, such as travel motivation, is provided elsewhere [45]. This study also reveals that online reviews could vary significantly across these platforms in terms of linguistic characteristics, semantic features, sentiment as well as impact on users of the websites (e.g., helpfulness). Fitness tracker apps, such as Strava or MapMyFitness, are primarily used to record an athlete's sports activities in a digital diary or to draft training plans. However, these apps also have a social dimension, e.g., by allowing to share trips or photos with peers [46]. Numerous studies compare trip data from different fitness tracker apps, with the conclusion that the popularity of road network segments can be best assessed through a combination of data from different platforms [47].

GIScience has employed social media data in a variety of studies [48], for instance, by exploring ways of acquiring geospatial information from social media [49]. A systematic literature review of 690 papers in 20 social media platforms that compares methods for location extraction from social media [50] reveals that the use of social media platforms is heavily skewed towards Twitter (54.2% of papers) and Flickr (20.3% of papers) with many alternative social media platform being barely used for this purpose. It also shows that social media information has been extracted in the areas of tourism and recreation (27.2%), crisis and disaster management (12.6%), transport (9.2%), and health (8.1%), among others. Social media data is nowadays a vital source of Volunteered Geographic Information (VGI), and due to bias from social media users, data integrity is a big concern. Therefore, various quality assurance processes for VGI have been proposed including peer reviews and automated error checking [51].

## 3. Data and Methods

### 3.1. Data from Social Media Apps

A total of 110 social media apps were examined for this study (Table 1). We grouped them into the following nine thematic categories: general purpose, lifestyle, photo sharing, blogging, video sharing, business, reunion networks, gaming, and travel. The initial list of social media apps and categories were adapted from two sources [52,53]. Some of the apps listed in these sources were removed from the analysis since they were no longer active, namely 43Things, Delicious, Fotolog, Friendster, Kiwibox, Google+, Path, Sprybirds, StumbleUpon, TravBuddy, Tournac, Tout, Uplike, and Vine. New apps that were recently launched, such as Tiktok, Bumble, and Medium, were added to the list. Apps with an API are marked with an asterisk (*) in Table 1. Since the geographic prominence of an app can be

influenced by the country in which it was launched, the table also provides the latter information for the listed apps. If not annotated differently, the default country where an app was launched is the United States.

**Table 1.** Social media apps categorized by theme together with the country of the app launch (AR … Argentina, CA … Canada, CN … China, DE … Germany, FR … France, IL … Israel, JP … Japan, KR … Korea, LU … Luxembourg, RU … Russia, TW … Taiwan, UK … United Kingdom), where * indicates availability of an API.

| General Purpose Social Networks | | | | | |
|---|---|---|---|---|---|
| AngelList | Gab | Facebook Messenger * | QQ * CN | Skyrock * FR | Twoo |
| Badoo * UK | GirlsAskGuys | Mixi * JP | Quora | Snapchat * | Vero * |
| Baidu Tieba * CN | Kickstarter * | Myspace | QZone | Spreely | Viber * LU |
| Discord * | LINE * KR | Nextdoor | Reddit * | Tagged | Vkontakte * RU |
| Douban | LinkedIn * | Pinboard * | Renren | Telegram * DE | WeChat * CN |
| Facebook * | MeetMe | Pinterest * | Sina Weibo * CN | The Dots | WhatsApp * |
| Foursquare * | Meetup * | ProductHunt * | Skype * | Twitter * | Yelp * |
| **Lifestyle** | | | | | |
| Academia.edu | CafeMom | English,baby! | Internations | PlentyofFish | Untappd * |
| ASmallWorld | Care2 | Flixster | Italki.com | Ravelry * | VampireFreaks |
| BlackPlanet | CaringBridge | Gaia | Last.fm * UK | ReverbNation | |
| Bumble | Crokes | Glocals | Minds * | SoundCloud * DE | |
| Busuu | Cross.tv | GoFundMe* | MyMFB | Taringa! * AR | |
| Buzznet | Doximity* | Goodreads* | Nexopia | Tinder | |
| **Photo sharing** | | | | | |
| 500px * CA | Dronestagram | Fotki | Instagram * | Pixabay * DE | Snapfish |
| DeviantArt * | Flickr * | Imgur * | Photobucket * | Shutterfly * | WeHeartIt |
| **Blogging** | | | | | |
| LiveJournal * RU | Medium * | Plurk * TW | Tumblr * | Xanga | |
| **Video sharing** | | | | | |
| | FunnyOrDie | Tiktok | YouTube * | YY | |
| **Business** | | | | | |
| eToro * IL | Ryze | Solaborate * | Viadeo * FR | XING * DE | |
| **Reunion networks** | | | | | |
| | 23andMe * | Ancestry.com | Classmates | MyHeritage * IL | |
| **Gaming** | | | | | |
| | Cellufun | Habbo | MocoSpace * | Zynga | |
| **Travel** | | | | | |
| | | CouchSurfing | Wayn | | |

### 3.2. Number of Related Research Papers

Three academic paper catalogs (Google Scholar, ACM Digital Library, WoS) were searched to (a) quantify research efforts related to the different social media platforms that provide an API and to (b) identify geospatial science topics covered in these research papers. Those three catalogs are commonly used for paper retrieval in review and meta-analysis studies [38,40,44].

Google Scholar is the most popular and largest search engine for scientific research books, articles, theses, abstracts, and court opinions from academic publishers, professional societies, online repositories, universities, and other web sites [54]. ACM Digital Library gives access to journals, conference proceedings, technical magazines, newsletters, books, and conference papers that cover a wide range of research themes but generally focus on computing [55]. WoS is a database that offers search capabilities for articles written back to 1900 and contains information gathered from thousands of scholarly journals, books, reports, and conferences [56]. It also has the "analyze

results" feature that can be used to get an overview of thematic categories of articles, conference papers, and books that are returned from a search related to a specific social media app. This "analyze results" feature provides further information about funding agencies, type of document published (articles, reviews, or proceedings paper), countries where the research was conducted, and the authors that have investigated the topic, among others.

For the search process, the names of the social media apps with APIs (57 apps in total) were used as search parameter one after another to retrieve papers whose title, abstract, or keyword sections contained those names. Only in ACM can one conduct searches by exclusively looking up title, abstract, or keywords. Google enables searches for titles or the entire article whereas in WoS one can use the "Topic" field, which looks up the title, abstract, and keyword. One challenge in this process is semantic search limitations, especially with homonyms. For example, social media app names that are also English words such as "Discord", "LINE", and "Medium" led to inflated results from all the catalogs and especially for Google Scholar. To reduce the number of irrelevant hits in search results, the search term was combined with the word "app" for "Medium", "LINE", and "Minds" for all three search catalogs.

### 3.3. Analysis Methods

This research involves a variety of qualitative and quantitative methods for the analysis of 110 social media apps that are related to the three research objectives mentioned earlier.

### 3.3.1. Presence and Characteristics of APIs

Social media data can be acquired through web scraping, data resellers, or APIs. Examining the presence of an API is critical as it eases the data collection and integration process. API documentation provides information about data that are accessible through different API endpoints. For this research, availability of spatial data was one of the aspects of interest examined.

Through online search and exploration of API directories such as Programmableweb [57], it was determined which of the 110 apps provide an API. These shortlisted apps were subsequently closely examined with regards to API characteristics, such as rate limits.

API rate limiting affects the amount of data that can be accessed within a given time or it limits the number of requests or calls a client can make, respectively. It is critical for scalability, security, and sustainment of an API.

The API documentation was also examined for presence and type of location data that were collected by the different social media apps. Geographic data provided through social media comes typically in the form of text (place names, addresses), geographic coordinates (latitude and longitude), or coordinates of the centroid of an administrative boundary (e.g., city, county, or country). This, in turn, determines the spatial resolution of the social media data set, which will subsequently affect the type of research that can be carried out.

An independent samples T-test was conducted to examine whether apps with a larger user base were more likely to provide API data access. Similarly, a second T-test was conducted for apps with APIs to explore the relationship between user numbers and the availability of spatial data through the API.

### 3.3.2. Academic Research Extent and Research Themes

Social media platforms have been the subject of numerous research studies across a wide range of disciplines. Whereas especially prominent social media apps appear to have attracted the analysis of its data and users, little is known about the role of less prominent apps on research activities. To narrow this research gap and to better understand the extent and current trends in research related to these platforms, two types of analysis are conducted. First, by using the names of social media apps with APIs (see Table 1) as search criteria on Google Scholar, ACM Digital Library, and WoS, the number of published documents, as counted by these online catalogs, is retrieved and discussed. An independent

samples T-Test is conducted for each of the three catalogs to examine whether a larger user base of a social media app is associated with a larger number of reported research documents. Second, to get an overview of the geospatial fields that apply social media analysis, the "analyze results" feature in the WoS platform was used for each social media app that comes with an API. This feature highlights different themes that the returned documents cover and illustrates, therefore, which social media app is associated with which discipline, including geospatial fields. The discovery of themes is based on title, abstract, and keywords of returned papers. The results of these two steps provide both quantitative and topical information about ongoing research related to a wide range of social media platforms with APIs. This information is meant to assist researchers with the identification of social media platforms that offer data related to their geospatial discipline (e.g., transportation) and in conveying a first idea about the abundance of research already conducted for each platform.

Furthermore, using title and abstract from each returned paper on WoS, a word cloud was created for selected categories of social media apps with APIs. This step highlighted terms commonly used in papers associated with different apps.

## 4. Results

### 4.1. User Statistics

The numbers of registered users for each of the 110 apps were retrieved through online searches. User statistics for the apps are summarized in Table 2, sorted in alphabetical order, together with year and month of the count. For 12 apps the number of registered users could not be determined. Facebook has the largest user base with over 2.4 billion users, followed by YouTube with 2 billion users. We hypothesize that user numbers can be used as a proxy for the availability of an API and the abundance of research literature found in relation to a social media app, which will be subsequently explored.

**Table 2.** User statistics.

| Social Media Sites with Approximate User Numbers | | |
|---|---|---|
| 23andMe-10 mio (April 2020) | GoFundMe-70 mio (April 2020) | ReverbNation-4 mio (2017) |
| 500px-15 mio (April 2020) | Goodreads-90 mio (April 2020) | Ryze-1 mio (April 2020) |
| Academia.edu-122 mio (April 2020) | Habbo-120 mio (2019) | Sina Weibo-497 mio (April 2020) |
| Ancestry.com-3 mio (April 2020) | Imgur-300 mio (April 2020) | Shutterfly-10 mio (2018) |
| AngelList-10 mio (2019) | Internations-3.6 mio (2019) | Skype-300 mio (2019) |
| ASmallWorld-28,500 (2018) | Instagram-1 bio (April 2020) | Skyrock-Unknown |
| Badoo-471 mio (April 2020) | Italki.com-5 mio (April 2020) | Snapchat-382 mio (April 2020) |
| Baidu Tieba (Postbar)-1.5 bio (2017) | Kickstarter-18 mio (April 2020) | Solaborate- Unknown |
| BlackPlanet- 15.8 mio (2007) | Last.fm-60 mio (April 2020) | SoundCloud-175 mio (April 2020) |
| Bumble-5.03 mio (2020) | LINE-600 mio (2017) | Snapfish-100 mio (April 2020) |
| Busuu-100 mio (2020) | LinkedIn-675 mio (April 2020) | Spreely-Unknown |
| Buzznet-Unknown | Live Journal-Unknown | Tagged-20 mio (2017) |
| Cafemom-146 mio (April 2020) | Medium-60 mio (2017) | Taringa!-30 mio (April 2020) |
| Care2-55 mio (April 2020) | MeetMe-Unknown | Telegram-200 mio (April 2020) |
| Cellufun-2 mio (2017) | Meetup-25.5 mio (2016) | The Dots-300,000 (2018) |
| Classmates-70 mio (2015) | Facebook Messenger-1.3 bio (April 2020) | Tiktok-800 mio (April 2020) |
| CaringBridge-500,000 (2013) | Mixi-14 mio (2013) | Tinder-50 mio (2018) |
| Crokes-46,000 (April 2020) | Minds-1.25 mio (2018) | Tumblr-23 mio (2019) |
| Cross.tv-743,829 (April 2020) | MocoSpace-2 mio (2017) | Twitter-340 mio (April 2020) |
| Couchsurfing-12 mio (April 2020) | MyMFB (LABYRA)-Unknown | Twoo-13 mio (April 2020) |
| DeviantArt-48 mio (April 2020) | MyHeritage-106 mio (April 2020) | Untappd-5 mio (2017) |
| Discord-250 mio (2020) | myspace-50 mio (2015) | VampireFreaks-2 Million (2009) |
| Dronestagram-30,000 (2016) | Nexopia-1.4 mio (2012) | Vero-3 mio (2018) |
| Douban-60 mio (2017) | Nextdoor-Unknown | Viadeo-50 mio (2017) |
| Doximity-1 mio (2017) | Photobucket-90 mio (April 2020) | Viber-1.1 bio (2019) |
| eToro-4.5 mio (2019) | Pinboard-22,000 (2013) | VKontakte (VK)-400 mio (2017) |
| English, baby!-1.6 mio (2012) | Pinterest-322 mio (2020) | Wayn-20 mio (2017) |
| Facebook-2.449 bio (2020) | Pixabay-Unknown | We Heart It-40 mio (2015) |
| Flickr-100 mio (April 2020) | PlentyofFish-150 mio (April 2020) | WeChat-1.151 bio (April 2020) |
| Flixster (Fandango)-67 mio (April 2020) | Plurk-100 mio (2013) | Whatsapp-1.6 bio (April 2020) |
| Fotki-Unknown | ProductHunt-600,000 (2018) | Xanga-27 mio (2006) |
| Foursquare-50 mio (April 2020) | QQ-731 mio (April 2020) | XING-10 mio (2016) |
| Funny or Die-2,830,000 (April 2020) | Quora-300 mio (2018) | Yelp-176 mio (2019) |
| Gab-200,000 (2017) | QZone-517 mio (April 2020) | YouTube-2 bio (April 2020) |
| Gaia Online-31 (April 2020) | Ravelry-9 mio (April 2020) | YY-Unknown |
| GirlsAskGuys-Unknown | Reddit-430 mio (April 2020) | Zynga-232 mio (2019) |
| Glocals-152,790 (April 2020) | Renren-31 mio (2018) | |

## 4.2. Availability of APIs and Spatial Data

APIs, where available, provide a gateway to social media app data either through a REST API or through GraphQL. The latter has recently become a more widely used alternative to REST APIs. Out of the 110 apps examined for this study, 57 apps provide an API (compare Table 1). Some apps provide multiple APIs for accessing different endpoints with specific datasets, as shown in Table 3.

**Table 3.** Characteristics of APIs and location data.

| Site | Rate Limits | Costs | API Name(s) | Location Data |
|---|---|---|---|---|
| 23andMe | Not specified | Free | 23andMe | Collected but not available through API |
| 500px | Pay to access | Paid | 500px | Available after paid subscription |
| Badoo | Not specified | Free | Badoo | Not specified |
| Baidu Tieba | Not specified | Not specified | Baidu Tieba | Not specified |
| DeviantArt | Varies | Free | DeviantArt & Sta.sh, oEmbed, RSS | Place name |
| Discord | Varies | Free | Discord | Place name |
| Doximity | 5000 requests per hour | Free | Doximity | Place name, coordinates |
| eToro | Not specified | Free | Discovery, Metadata, System, User | Place name |
| Facebook | Varies | Free | Graph, Marketing, Live Video, Pages | Place name, coordinates |
| Flickr | 3600 queries per hour | Free | Flickr Rest | Coordinates |
| Foursquare | Varies | Free and paid | Places | Place name, coordinates |
| Goodreads | Not specified | Free | Goodreads | Place name/query per location |
| GoFundMe | Not specified | Free | GoFundme | Place name |
| Imgur | 12,500 requests per day | Free | Imgur | Collected but not available through API |
| Instagram | Varies | Free | Basic Display, Graph | Coordinates |
| Kickstarter | Not specified | Free | Kickstarter Status | Not available |
| Last.fm | Varies | Free | Last.fm | Collected but not available through API (query per location) |
| LINE | Varies | Free and paid | Messaging | Place name, coordinates |
| LinkedIn | Varies | Free | LinkedIn | Place name |
| LiveJournal | Varies | Free | LiveJournal | Place name |
| Medium | Not specified | Free | Medium | Collected but not available through API |
| Meetup | Not specified | Free | Meetup Rest | Place name, coordinates |
| Facebook Messenger | Varies | Free | Attachment Upload, Broadcast, Customer Matching, Handover Protocol, ID Matching, Messenger Code, Messaging Insights, Messenger Profile, Personas, Send, Messaging Feature Review | Coordinates |
| Mixi | Not specified | Free | Graph -People, Groups, People lookup, Voice, Updates, Check, Photo, Message, Diary, Check-in, Profile Image, Persistence, Calendar | Place name, coordinates |

**Table 3.** *Cont.*

| Site | Rate Limits | Costs | API Name(s) | Location Data |
|------|-------------|-------|-------------|---------------|
| MocoSpace | Not specified | Free | MocoSpace | Not specified |
| Minds | Not specified | Free | Minds | Not specified |
| MyHeritage | Not specified | Free | Family Graph | Place name |
| Photobucket | Varies | Free and paid | Photobucket | Coordinates |
| Pinboard | 1 call per user every 3 s | Paid | Pinboard | Not specified |
| Pinterest | Approved app can make 100 calls/day | Free | Pinterest | Place name |
| Pixabay | 5000 requests per hour | Free | Pixabay | Collected but not available through API (query per location) |
| Plurk | Not specified | Free | Plurk | Place Name |
| ProductHunt | Varies | Free | ProductHunt GraphQL | Collected but not available through API |
| QQ | Not specified | Free | User Information, Relationship Chain, Application promotion, Payment | Not specified |
| Ravelry | Not specified | Free | Ravelry | Collected but not available through API |
| Reddit | 60 requests per minute | Free | Reddit | Place name |
| Sina Weibo | Varies | Free | Sina Weibo Rest, Search, Geo, Test | Place name, coordinates |
| Shutterfly | Not specified | Not specified | Shutterfly | Collected but not available through API |
| Skype | Not specified | Free | Skype | Place name |
| Skyrock | Varies | Free | Skyrock | Place name, coordinates |
| Solaborate | Not specified | Free | Solaborate | Not specified |
| SoundCloud | Varies | Free | SoundCloud | Place name, coordinates |
| Snapchat | Not specified | Free | Snap Kit | Collected but not available through API |
| Taringa! | Not specified | Free | Taringa! Rest | Not specified |
| Telegram | Varies | Free | Telegram Rest, Bot | Place name |
| Tumblr | Varies | Free | Tumblr | Place name |
| Twitter | Varies | Free and paid | Ads, Search Tweets, Streaming, and Direct Message. | Place name, coordinates |
| Untappd | 100 calls per key | Free | Untappd | Place name, coordinates |
| Vero | Not specified | Not specified | Vero | Collected but not available through API |
| Viadeo | Last 50 items of endpoints | Free | Viadeo | Collected but not available through API |
| Viber | Varies | Free | Viber | Place name, coordinates |
| VKontakte | Varies | Free | VKontakte | Place name |
| WeChat | Not specified | Not specified | WeChat | Place name, coordinates |
| WhatsApp | Varies | Paid | WhatsApp Business | Not specified |
| XING | Varies | Not specified | XING E-Recruiting | Place name |
| Yelp | 5000 API calls per day | Free | Fusion | Place name, coordinates |
| YouTube | Varies | Free | Iframe Player, Data, Analytics, Live Streaming, Reporting | Place name |

Apps with large user numbers are more likely to provide an API than those with smaller user numbers. An independent samples T-test showed that the 53 out 57 apps with an API and known user counts had higher user numbers (Mean (M) = 345,582,491, Standard Deviation (SD) = 540,839,941) compared to the 53 apps without an API (M = 70,684,659, SD = 146,634,689), t(df) = 96, $p < 0.001$. Furthermore, among those apps with an API, an independent samples T-test showed that the mean number of users was higher for apps that provided spatial data in their API (M = 395,848,485, SD = 574,334,200) than for those apps that did not (M = 262,643,600, SD = 483,078,816), t(df) = 51, $p = 0.015$.

All APIs require users to register their apps to obtain API access credentials, such as an ID number and API key. API access requests for most apps were immediately approved, but for apps, such as Untappd or Facebook, this process takes some time.

Rate limiting methods were applied to numerous APIs (Table 3). For a few apps, such as DeviantArt, Discord, Last.fm, and LINE, which do not have a set rate limit, an effective rate limit varies depending on the frequency at which clients make requests and its effect of the app stability. For others, such as 23 and Me, eToro, and Goodreads, rate limitations were not disclosed. Doximity, Flickr, Imgur, Pinboard, and Pinterest have fixed API request rate limits. Besides setting rate limiting per given time, rate limits can also be implemented through fees. Most of the apps provide free access or a mix of free and paid subscriptions. For example, Twitter charges for access to its Premium and Enterprise APIs, whereas the Search and Streaming APIs are free of charge. Foursquare and Photobucket are free for noncommercial use but charge fees for commercial use of their APIs. As opposed to this, WhatsApp Business, Pinboard, and 500px APIs provide access through paid subscriptions only.

The apps that collect and share spatial data provide different types of location information, such as coordinates of the centroid of an administrative boundary. Based on information from the terms of service and privacy policies of various apps, location data was not provided in the APIs of a few apps even though it is collected in the background using IP addresses. In summary, 22 apps did not provide access to spatial data while 35 provided place name, geographic coordinates, or both through the API according to their API documentation (Table 3). Some APIs, such as that of Last.fm, allowed only running spatial queries, such as using the geo.getTopTracks or geo.getTopArtists API methods to query these endpoints using country names. Location data from several apps stems from user profiles only while a few apps, such as Twitter, Facebook, and Instagram provide spatial data affiliated with multimedia, such as pictures, videos, or text. 500px provides this information only once a paid subscription is made. The Flickr API is the only one that reports the accuracy of the coordinates it provides.

Although Table 3 provides information about rate limits, costs, and location data, additional limitations as to whose data can be downloaded may apply. In some cases, only one's own data can be accessed through the API, or those from other users who gave specific permission as specified, for example, for the Facebook Graph API or the WeChat API [58,59].

### 4.3. Research Papers on Google Scholar, ACM Digital Library, and Web of Science

Apps such as Facebook, Instagram, Twitter, and YouTube had the highest numbers of papers published about them (Table 4), indicating that they are prominent platforms within the research community. These are also the apps with high user numbers. To quantify these results in more detail, first, the median of user numbers across social media apps in Table 2 (where available) was computed as 100,000,000. Next, each app was assigned to one of two groups, namely either the one group that contained apps with user numbers below the median or the other group with user numbers above the median. An independent samples T-test showed that in Google Scholar the apps falling into the upper user group had more paper counts (M = 331,516.9, SD = 543,184.4) compared to those in the lower user group (M = 75,528.58, SD = 227,738.4), t(df) = 49, $p < 0.031$. Similarly, in the ACM Digital Library the apps in the upper user group had more paper counts (M = 3766.5, SD = 6928.3) than those in the lower user group (M = 283.9, SD = 640.8), t(df) = 49, $p < 0.014$. Furthermore, in the WoS the

apps in the upper user group had more paper counts (M = 3220.3, SD = 8115.4) than those in the lower user group (M = 139.8, SD = 320.5), t(df) = 49, *p* = 0.059. In the case of WoS, the level of significance for the difference in means is slightly above 5%, whereas for the other two search catalogs it is below that value.

**Table 4.** Number of research papers associated with the different social media platforms.

| Site | Google | ACM | WoS | Site | Google | ACM | WoS |
|---|---|---|---|---|---|---|---|
| 23andMe | 9590 | 50 | 656 | Pinterest | 274,000 | 714 | 214 |
| 500px | 2910 | 36 | 4 | Pixabay | 25,400 | 36 | 24 |
| Badoo | 4780 | 33 | 17 | Plurk | 15,600 | 37 | 16 |
| Baidu Tieba | 1680 | 1273 | 8 | ProductHunt | 332 | 0 | 0 |
| DeviantArt | 5560 | 72 | 12 | QQ | 1,010,000 | 2432 | 37503 |
| Discord | 33 | 4 | 4095 | Ravelry | 889 | 30 | 0 |
| Doximity | 1090 | 1 | 68 | Reddit | 307,000 | 1353 | 267 |
| eToro | 1280 | 2 | 7 | Shutterfly | 1140 | 32 | 6 |
| Facebook | 1,830,000 | 22,108 | 15,663 | Sina Weibo | 32,100 | 1430 | 392 |
| Facebook Messenger | 291,000 | 24,144 | 78 | Skype | 76,700 | 3333 | 918 |
| Flickr | 139,000 | 5485 | 924 | Skyrock | 4420 | 8 | 3 |
| Foursquare | 23,000 | 1846 | 345 | Snapchat | 18,500 | 659 | 234 |
| Goodreads | 14,900 | 99 | 46 | Solaborate | 9 | 0 | 0 |
| GoFundMe | 3240 | 29 | 29 | SoundCloud | 138,800 | 154 | 20 |
| Imgur | 4200 | 58 | 9 | Taringa! | 7860 | 2 | 204 |
| Instagram | 194,000 | 2503 | 1492 | Telegram | 68,000 | 342 | 312 |
| Kickstarter | 15,500 | 310 | 186 | Tumblr | 229,000 | 563 | 183 |
| Last.fm | 15,000 | 826 | 102 | Twitter | 1,700,000 | 17235 | 12111 |
| LINE | 1330 | 23 | 11 | Untappd | 232 | 3 | 0 |
| LinkedIn | 525,000 | 2998 | 603 | Vero | 556,000 | 125 | 8 |
| LiveJournal | 24,100 | 905 | 73 | Viadeo | 3590 | 4 | 3 |
| Medium | 411 | 3 | 2 | Viber | 15,600 | 149 | 41 |
| Meetup | 11,600 | 326 | 50 | VKontakte | 16,000 | 69 | 95 |
| Minds | 10 | 0 | 158 | WeChat | 345,000 | 707 | 731 |
| Mixi | 12,400 | 129 | 11 | WhatsApp | 39,600 | 1347 | 954 |
| Mocospace | 179 | 3 | 0 | XING | 1,040,000 | 2773 | 1540 |
| MyHeritage | 14,100 | 9 | 10 | Yelp | 20,500 | 1342 | 349 |
| Photobucket | 5210 | 71 | 2 | YouTube | 1,360,000 | 9744 | 4381 |
| Pinboard | 2820 | 47 | 14 | | | | |

Minds, Solaborate, Kickstarter, and Mocospace APIs were among the apps with the lowest totals across the three catalogs when queried for papers. "Discord", "LINE", "Medium", which exist as English words, resulted in many irrelevant hits. XING, which is an app and a person's name, led to unrelated results as the catalogs also tallied papers that were written by authors with that name. The latter problem could be mitigated by excluding author name in the search criteria from WoS whose results were also used to create the word cloud. Mocospace, ProductHunt, Ravelry, Solaborate, and Untappd did not have any papers affiliated with these social media platforms on WoS.

Table 4 reveals strong discrepancies in the number of returned papers between the three search portals. For each social media app, except for two platforms whose names also resemble an English term (Discord, Minds), Google Scholar by far outnumbers ACM and WoS search results. Google Scholar is one of the most widely used tools for researchers to search scientific information, and through parsing the entire Web its coverage is much more extensive than other multidisciplinary commercial databases like WoS or Scopus [60]. However, only up to 1000 search results can be displayed on the Google Scholar Website, and no API is available for additional search. Results from an earlier study show that 55% of the more than 2.6 million sample documents covered by WoS from 2009 and 2014 were freely available in some form from Google Scholar, e.g., as open access documents from publisher Websites or through links to other repositories, such as researchgate or arxiv [61]. ACM displays up to 2000 search results, where filters can be used to specify the subset of records to be displayed in cases where more than 2000 documents are returned upon a query. As opposed to this, the WoS allows to view all returned records by browsing through the result pages.

*4.4. Research Themes*

Research themes for papers returned from the WoS search were extracted using the "analyze results" feature. Out of the 57 apps with an API, 26 apps had spatial themes (Geography, Remote Sensing, Transportation, and Urban Planning). The frequency of the geospatial themes as found in the 3095 papers that are associated with those 26 apps are shown in Figure 1. The bars are overlaid with the number of papers that were found to be associated with each spatial theme for the different apps. The apps are sorted in descending order of the total number of papers found to be associated with a spatial theme, starting with XING on top (1113 papers), followed by Twitter (674 papers) and QQ (532 papers). Several apps had only one paper with a spatial theme, such as Tumblr or Reddit. The dominating theme is Geography, which can be found in papers for all apps listed in the chart. The other spatial themes are less frequently discussed in research outlets, i.e., Transportation for 73% of the apps, Remote Sensing for 62%, and Urban Planning for 69%. Common non-spatial themes found in papers related to apps with APIs include Materials science, Chemistry, Physics applied, Biochemistry molecular biology, Engineering electrical electronic, Oncology, Nanoscience, Communication, Computer science information systems, and Environmental Sciences. These themes are not listed in Figure 1 though.

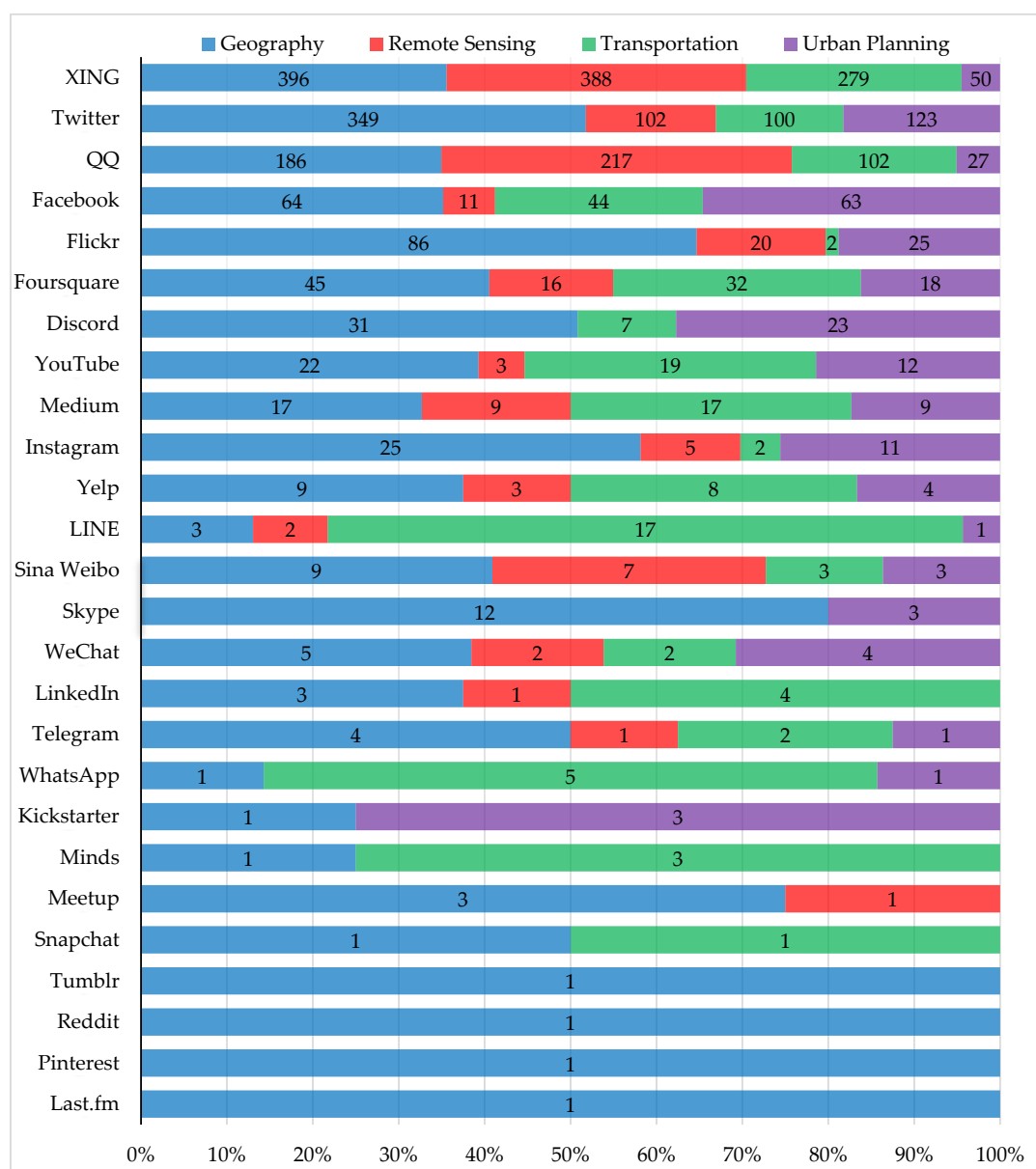

**Figure 1.** Spatial themes of papers associated with social media apps.

The overview in Figure 1 adds to the existing body of knowledge by provision of topical research relevance for wide range of social media apps. As opposed to this, several earlier studies analyzed in-depth research activities associated with individual social media apps one at a time, such as Twitter [44] or Facebook [42]. These are, as can be seen in Figure 1, among the most widely analyzed social media platforms.

Figure 2 shows word clouds formed by papers from WoS that are associated with social media apps that provide APIs, where apps are grouped by theme according to Table 1. No word cloud was created for the gaming category because Mocospace, which is the only app with an API in that category, did not yield any papers from WoS. The key terms for each word cloud generally align well with the theme that the set of corresponding apps was assigned to in Table 1. For example, papers on blogging revolve around networking, social media, health, and community. Business app-related papers discuss trading, investment, decision making, and risk, whereas papers on photo sharing networks mention images, Instagram, Flickr, photo, or visual. The word clouds, therefore, demonstrate the different facets of activities and concepts that research papers discussing these apps cover in their analyses.

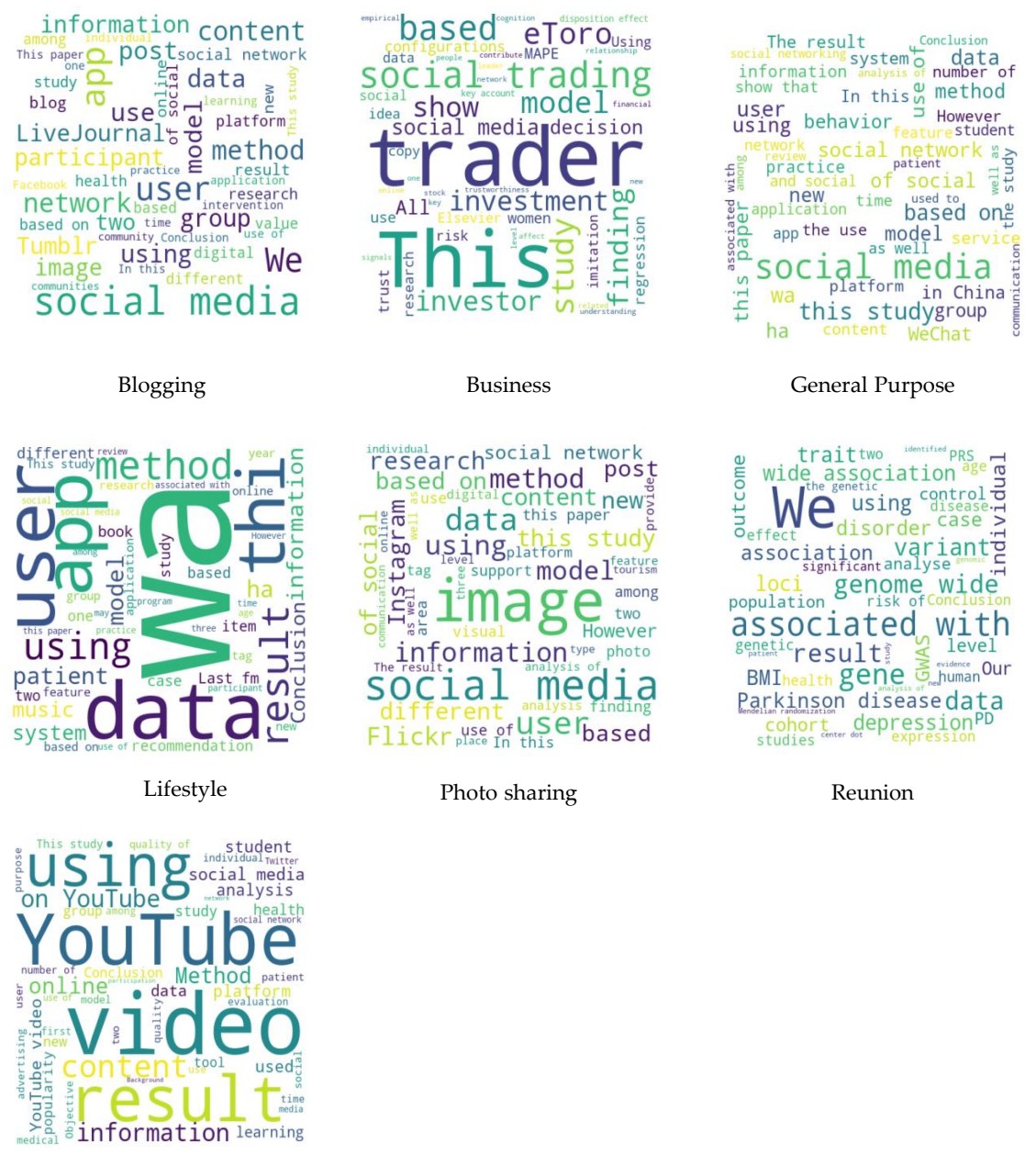

**Figure 2.** Word clouds for different categories of social media apps.

## 5. Discussion

According to Statista, more than 50% of the world's population use social media [62]. This demonstrates the important role of social media apps today. Communication and user profile data as well as the user-generated content shared on these platforms provide a valuable resource for researchers across numerous disciplines. The user statistics summarized in this study give a good indication of the popularity of the different apps. Prominence of social apps, as it can be found, for example, for Facebook, Instagram, and YouTube, also comes with increased interest in the research community, as reflected in numerous research papers related to particularly prominent social media apps.

To stimulate innovation and help expand their business services, about half of the analyzed social media platforms give developers access to some of their data through APIs. This trend began in the early 2000s with e-commerce companies such as eBay and Amazon leading these efforts [63]. Statistical

tests revealed that social media apps with a large user base also tend to provide APIs more often compared to those apps with lower user numbers. This is because development and support of an API require certain developer resources and infrastructure. In turn, provision of an API access may also lead to growth in user numbers as is the case with e-commerce platforms such as eBay because third-party developers are able to embed API functions onto their websites. In other words, companies behind apps with large user numbers have financial resources to develop and maintain APIs. Investment in an API pays off in the form of increased platform visibility and fees charged for the use of such an API for business services. For instance, Twitter as of 2011 got 15 billion calls a day on their site, three-quarter of which came through their API, and Facebook received 5 billion calls a day through its APIs [64]. All the APIs reviewed in this study required filling out an application form. For most apps, the approval process occurs almost immediately. However, for some apps, such as Untappd, the process can take up to three weeks [65] and for Instagram it can take one week or more [66]. The rigorous application process is aimed at tracking how the data acquired from these apps are used as users are asked to specify data use intentions during the application process [67]. 500 px and Pinboard are some of the apps where API users are charged a fee whereas for other apps API use was partially or entirely free. Foursquare and Twitter APIs are operated in a freemium type of model that offers basic data access for free for noncommercial use, but increased data access or commercial use are charged for. For instance, Foursquare provides twice the amount of data on the paid subscription of its API, and the premium and enterprise versions of the Twitter API give access to the full archive of tweets.

API rate limits express the allowable request per time interval and help with the stability and security of an app. Rate limiting techniques include maximum allowable requests per given time span, which, when exceeded, leads to the API calls being disconnected or the bandwidth being reduced. The Doximity API, for instance, allows its users to make 5000 API requests per hour whereas Flickr allows 3600 queries per hour. Twitter and Viber rate limits depend on the information being requested or the API call being made. Rate limits control the volume of data a developer can have access to at a given time. Access to data is also affected by a user's privacy settings. The Facebook Graph API rate limit varies in that it implements a user level rate limiting per access token, whereas the app level limits are based on counts of the app ID being used. API level limits are based on number of API calls per second and IP level rate-limiting is based on the number of calls assigned to a particular IP address [68].

Since social media apps gather user data, in recent years, a variety of government-mandated policy changes regarding data privacy were initiated. An example is Europe's GDPR, which aims at giving the users more control over how their data is collected, processed, stored, and transferred in the 28 member states of the European Union [69]. This has led to some social media apps restricting data access through their APIs as they changed their privacy policies to abide by the new regulation. This, in turn, affected data accessibility through the APIs. Facebook, for instance, rolled out new API restrictions in the wake of the Cambridge Analytica data scandal and GDPR which were aimed to close loopholes in the website privacy policies [36]. 500 px announced in 2018 that it will only provide paid access to its API in order to protect its data and improve its site performance [70]. Such changes in policies tend to further reduce the availability of social media app data for geospatial research because location data is widely considered personal or private data.

Location data is the backbone of geospatial research. APIs provide this data in the form of place names, geographic coordinates, address, time zone, or IP address, among others. Location data on social media can be extracted from message context or metadata, user profiles, the social network of friends and connections, tags, multimedia content (photos, videos), and links to other connected social media pages [50]. The main location types of data found on APIs in this study were place names and geographic coordinates, whereas some apps also collected IP address information. In some apps, location data is not made available through the API. For example, 23andMe, Imgur, and Last.fm collect location data from IP addresses but do not provide access to this data on their respective APIs. Additional restrictions hinder access to comprehensive location data through an API. When a user's privacy settings do not allow sharing of this data then the information will not be retrievable.

For example, the Facebook Graph API prevents the retrieval of data of a particular user due to their privacy settings [58]. In addition, a low percentage of social media content has been geotagged with geographic coordinates, for instance, only about 1-2% of tweets are geotagged [71]. The integrity of the location information extracted from social media can be questionable in some aspects as studies have found that some users provide incorrect locations intentionally in their user profiles [72].

Social media data has boosted research in various areas, which becomes evident from the number of papers retrieved from Google Scholar, the ACM Digital Library, and the WoS. The research themes on Figure 1, based on the papers retrieved from the WoS, illustrate the geospatial disciplines that have been researched in relation to the different apps. Among all 89,839 papers retrieved from the WoS for the 57 apps considered that provide an API, only 3095 are associated with geospatial disciplines. These 3095 papers are related to 26 out of the 57 apps. This low number of papers relative to the total of 89,839 papers reveals that the role of geospatial research within the entire social media research arena is a relatively small one. Instead, research in other disciplines, such as communication or computer science, more often report on social media data analysis.

There is a tendency that only a handful of apps are used for geospatial research even though other lesser known apps also provide datasets that could potentially drive research and reveal insights about various phenomena [50]. A total of 17 apps out of the 57 apps with APIs can in this study be identified as potential data sources for future research as they provide geographic data either from media or user profile information and come only with a small number of research papers published so far. These apps include DeviantArt, Discord, Doximity, eToro, Goodreads, Last.fm, LINE, LiveJournal, Mixi, MyHeritage, Photobucket, Plurk, Sina Weibo, Skyrock, Untappd, Viber, and Vkontakte. As an example of a successful attempt to retrieve geospatial information, Figure 3 depicts the distribution of location data obtained from the Plurk and DeviantArt APIs based on user profiles. The Plurk API facilitates batch processes such as adding friends or fans and getting bookmarks. The API can be also be used to look for search terms of interest which returns a JavaScript Object Notation (JSON) response with only the 20 latest plurks. The JSON report includes place names associated with the user profile locations. The red markers in Figure 3 show the profile locations of users who included the "Coronavirus" search term in their plurks. This data was accessed through the/APP/PlurkSearch/search endpoint. The DeviantArt API allows one to get the country names of watchers who are followers or fans of a DeviantArt user's page. Using the/user/watchers/{username} endpoint facilitated the extraction of the location, i.e., the country name, of about 100 watchers of a user with the name boldfrontiers (black markers in Figure 3).

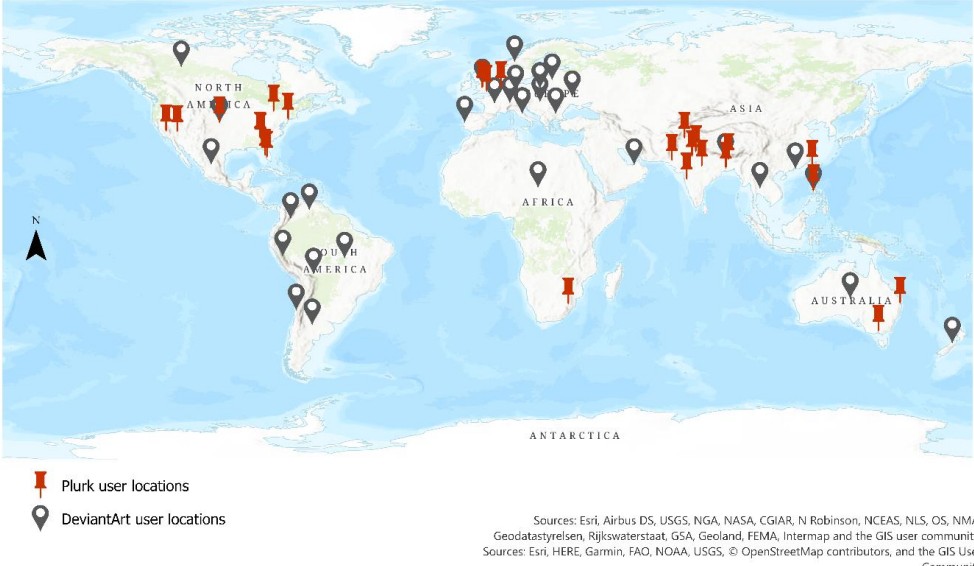

**Figure 3.** Location data from Plurk and DeviantArt.

## 6. Conclusions

The overall goal of this study was to explore the potential of less known social media apps for geospatial research by evaluating 110 social media apps, among which more than half were found to provide APIs. APIs give developers data access and facilitate examination of the characteristics of data availed, including the provision of spatial data. As learned from the different API documentations, all the APIs considered store location data from their users in one form or another, but this information may not be available due to government regulations, app privacy policies, or a user's privacy settings.

Recent governmental policy guidelines, such as GDPR and CCPA, have affected social media data access as they led to stricter rules in a wide range of social media apps. It will be of interest to investigate how this impacts data access and consequently research that relies on social media data in the long run. Facebook, which is one of the biggest social media platforms, introduced restrictions that curtail full data access in April of 2018 [36] as a result of GDPR. Besides this, another common obstacle in data extraction from APIs is frequent changes of policies, access regulations, functionality, or user registration and approval procedures, as well as outdated or missing API documentation. The latter necessitates substantial and time-consuming exploration, code testing, and internet search to obtain successful data access.

Among the list of 35 APIs that provide spatial data in form of place names or geographic coordinates, the spatial granularity of available data varies and, therefore, influences the type of research that it can be applied for. Spatial accuracy information from apps that provide coordinates is only given within the Flickr API. Aspects of future work include, therefore, the assessment of the quality of spatial data accessible in the 17 apps that provide location data through an API, even though they are rarely featured in geospatial research.

Social media sites collect an ever-increasing amount of data from their users, but privacy concerns may render access to these datasets impossible unless it is anonymized, and personal information is removed. Even if so, ethical use of the data is required. Therefore, the future may bring more rigorous approval procedures of API access requests with even follow-up audits to ensure that data is being used as intended. Future work will closely examine the consequences of such anticipated changes in API regulations on the social media analysis.

**Author Contributions:** Conceptualization, Hartwig H. Hochmair and Innocensia Owuor; methodology, Innocensia Owuor and Hartwig H. Hochmair; software, Innocensia Owuor; validation, Innocensia Owuor and Hartwig H. Hochmair; formal analysis, Innocensia Owuor; data curation, Innocensia Owuor; funding acquisition, Hartwig H. Hochmair; project administration, Hartwig H. Hochmair; resources, Hartwig H. Hochmair; supervision, Hartwig H. Hochmair; visualization, Innocensia Owuor; writing—original draft, Innocensia Owuor; writing—review and editing, Innocensia Owuor and Hartwig H. Hochmair. All authors have read and agreed to the published version of the manuscript.

**Funding:** We acknowledge financial support through a UF-CALS matching assistantship granted to the first author.

**Acknowledgments:** The authors thank two anonymous reviewers for their constructive feedback.

**Conflicts of Interest:** The authors declare no conflict of interest.

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
