# Peer review of "An Overview of Social Media Apps and their Potential Role in Geospatial Research"

_ijgi, doi:10.3390/ijgi9090526_

Round 1
Reviewer 1 Report
The main purpose aim of the paper is to explore which applications provide access to location data and be a useful resource for geospatial research.
I can say that the paper is interesting and it has potential. In general, I can say that it is well written, it has a good structure. The authors present their work in a concise and comprehensive way and the writing has a good flow. However, it needs improvements and clarifications.
First, starting from the abstract, I can say that it needs to present the main results of the study and also highlight the findings. For example, the phrase “The study concludes that among the large number of social media apps available today indeed a few of the less known apps are worthwhile closer examination as they might provide access to geospatial data for further analysis in addition to data from the big players in that research arena.” should be associated with numeric results.
The introduction needs to explain in better detail the aims of the paper, and the motivation of the authors to perform this work. In addition, I feel that it is important to highlight out more explicitly the contributions of the work and how it could be assistive to the related research community.
The paper also needs to present additional related works and explain how it builds on the findings of related works and approaches. Also I suggest to present in better detail for each of the related work i) the main aims of it and ii) the main results and findings.
The rational and the criterial of the examination of each one app could be stated more explicitly in the start of Section3.
Sub-section 3.3 that is the Analysis Methods is a core part of the authors work. I feel that it needs to explain in better detail the importance of each analysis part, the necessity for examination as well as needs to have a better connection with the results (for example what the user statistic information in table 2 offers to the study).
Please extend and explain the discussion of the results. Please add a paragraph after each table to make a deeper comparative discussion on the results of the apps.
I can say that the results are interesting but a better and deeper discussion needs to be made. The results in table 4 need also further discussion. The same stands for the results presented in Figure 1.
The paper will benefit from a careful proofreading. For example, Line 304 starts in a ‘bizarre’ way. Please have the paper proofread again.
Author Response
Dear reviewer,
Thank you for providing us with constructive feedback and comments. Most of the reviews ask for additional explanations and further discussion of results. These additions are highlighted in yellow in the revised manuscript. Please see below how we addressed reviewer comments in the revision.
REVIEW 1:
Comments and Suggestions for Authors
The main purpose aim of the paper is to explore which applications provide access to location data and be a useful resource for geospatial research. I can say that the paper is interesting and it has potential. In general, I can say that it is well written, it has a good structure. The authors present their work in a concise and comprehensive way and the writing has a good flow. However, it needs improvements and clarifications.
Point 1: First, starting from the abstract, I can say that it needs to present the main results of the study and highlight the findings. For example, the phrase “The study concludes that among the large number of social media apps available today indeed a few of the less known apps are worthwhile closer examination as they might provide access to geospatial data for further analysis in addition to data from the big players in that research arena.” should be associated with numeric results.
Response:
The abstract has been extended with the presentation of results from the three objectives, including numerical findings.
It states that out of the 110 examined applications 57 were found to have APIs. It also presents the mean number of research papers found on Google Scholar, the Association for Computing Machinery (ACM) digital library and Web of Science (WoS) for the different social media apps analysed, and explains that for 26 apps at least some papers evolve around a geo-spatial discipline, such as Geography, Remote Sensing, Transportation, or Urban Planning.
Finally, it states that 17 applications were identified to be potentially useful for geospatial research since the apps are quite underexplored in academia even though they provide valuable geospatial data.
Point 2: The introduction needs to explain in better detail the aims of the paper, and the motivation of the authors to perform this work. In addition, I feel that it is important to highlight out more explicitly the contributions of the work and how it could be assistive to the related research community.
Response:
The introduction provides now three bulleted items detailing the research objectives. This is followed by the description of the overarching research goal of this study.
It is hoped that the research community will benefit from information provided about the different apps with their APIs to identify specific apps that could help researchers tackle previously underexplored research areas.
More aspects of user selection bias are now presented in the introduction, substantiating the presumption that the joint analysis of data from multiple social media platforms can help mitigate biases in data analysis. A few sentences about fitness tracker apps where also added, pointing out that data from a single tracker app will cause bias and joint analysis can have certain advantages.
Point 3: The paper also needs to present additional related works and explain how it builds on the findings of related works and approaches. Also, I suggest presenting in better detail for each of the related work i) the main aims of it and ii) the main results and findings.
Response:
The most relevant previous work is in our opinion those studies which review literature and research on prominent social media applications (e.g. Twitter) or summarize their use for certain tasks or in certain contexts (e.g. in academic environments). Our research adds to the existing body of knowledge by provision of topical research relevance for a wider range of social media apps. This has been added to section 4.4.
Previous research also discusses the important role of APIs for social media analytics. Our study describes which social media apps come with an API, and what the characteristics of these APIs are for each app. We added several new references to API documentations (highlighted in the reference section) when we discussed the features of the analysed apps. In addition to this, our research shows the statistical relationship between the size of the user base of an app and the availability of an API.
In the revision we added now statistical tests (section 4.3) which demonstrate the relationship between the size of the user base of an app and the number of research papers published (as counted in Google Scholar, ACM, or Web of Science). Whereas previous research been mostly analysed with regard to a single social media platform at a time, the relationship between app prominence and number of research documents is another new contribution of our study. It demonstrates that smaller apps are under represented in the research arena and thus have a large potential for further exploration, as described in the discussion section.
Point 4: The rational and the criterial of the examination of each one app could be stated more explicitly in the start of Section3.
Point 5: Sub-section 3.3 that is the Analysis Methods is a core part of the authors work. I feel that it needs to explain in better detail the importance of each analysis part, the necessity for examination as well as needs to have a better connection with the results (for example what the user statistic information in table 2 offers to the study).
Response:
We added further details to section 3.3., particularly section 3.3.1 and 3.3.2, to respond to the two previous points mentioned by the reviewer. We added the explanation for table 2 that the user numbers of an app shown in the table can be used as a proxy for the availability of APIs, and for the abundance of research conducted on an app. This was confirmed using a series of independent samples T-tests.
Each table presented is closely tied to one or several of the three research objectives (i.e., provision of APIs, research published on each app, and research topics associated with each app). We added now the country where an app was launched for each app in Table 1, following the newly added statement in the introduction section that this has a strong impact on app use in a country, and therefore on user selection bias.
Point 6: Please extend and explain the discussion of the results. Please add a paragraph after each table to make a deeper comparative discussion on the results of the apps.
Response:
We added additional descriptions when we felt that further discussion of the results would be helpful.
We expanded the discussion section to better tie our findings to existing research and to provide additional details about our findings, especially with respect to APIs. A few references were added as well.
Point 7: I can say that the results are interesting but a better and deeper discussion needs to be made. The results in table 4 need also further discussion. The same stands for the results presented in Figure 1.
Response:
We added a paragraph highlighting the findings on Table 4 and providing further explanations on the academic search catalogues that were used in this study.
Results of Figure 1 are discussed in multiple parts of the manuscript (section 4.4., conclusions), and a paragraph was added to section 4.4. to point out our novel contribution.
Point 8: The paper will benefit from a careful proofreading. For example, Line 304 starts in a ‘bizarre’ way. Please have the paper proofread again.
Response 8:
We cannot see the line numbers, so we cannot specifically address the sentence the reviewer mentions, but we carefully proof-read the manuscript.

Reviewer 2 Report
Author propose a review of 110 social media applications that can be use in geo-spatial research. The review is well written and provide very interesting information about such social media applications. Conclusions should include some information about authors future research on the topic.
Author Response
Dear reviewer,
Thank you for providing us with constructive feedback and comments. Most of the reviews ask for additional explanations and further discussion of results. These additions are highlighted in yellow in the revised manuscript. Please see below how we addressed reviewer comments in the revision.
REVIEW 2:
Point 1: Author propose a review of 110 social media applications that can be use in geo-spatial research. The review is well written and provide very interesting information about such social media applications. Conclusions should include some information about authors future research on the topic.
Response:
A few potential directions for future work that have emerged as a result of this study were now added to the conclusions.
This includes an assessment of how new privacy regulations will impact data access through APIs and research on social media data in general. More details on new privacy regulations were also added to the discussion section.
The conclusions mention as future work the need to investigate the quality of spatial data that can be gathered from the APIs of less known apps that were identified in the study.

Round 2
Reviewer 1 Report
Authors addressed the previous comments and the paper has been substantially improved compared to the original submission.
I vote for the acceptance of the article.